# Deployed Veterans exhibit distinct respiratory patterns and greater dyspnea during maximal cardiopulmonary exercise: A case-control study

Thomas Alexander[1], Matthew A. Watson[1], Jacquelyn C. Klein-Adams[1], Duncan S. Ndirangu[1], Jorge M. Serrador[2], Michael J. Falvo[1,2,3‡], Jacob B. Lindheimer[4,5‡]*

1 VA Airborne Hazards and Burn Pits Center of Excellence, VA New Jersey Health Care System, East Orange, New Jersey, United States of America, 2 Department of Pharmacology, Physiology and Neuroscience, New Jersey Medical School, Rutgers – The State University of New Jersey, Newark, New Jersey, United States of America, 3 Department of Physical Medicine and Rehabilitation, New Jersey Medical School, Rutgers – The State University of New Jersey, Newark, New Jersey, United States of America, 4 William S. Middleton Memorial Veterans Hospital, Madison, Wisconsin, United States of America, 5 Department of Kinesiology, University of Wisconsin-Madison, Madison, Wisconsin, United States of America

‡ These authors contributed equally to this work as senior author

* Jacob.Lindheimer@va.gov

**Data Availability Statement:** All data were collected at the Department of Veterans Affairs and the signed subject consent forms and HIPAA

## Abstract

### Background

Exertional dyspnea and exercise intolerance are frequently endorsed in Veterans of post 9/11 conflicts in Southwest Asia (SWA). Studying the dynamic behavior of ventilation during exercise may provide mechanistic insight into these symptoms. Using maximal cardiopulmonary exercise testing (CPET) to experimentally induce exertional symptoms, we aimed to identify potential physiological differences between deployed Veterans and non-deployed controls.

### Materials and methods

Deployed (n = 31) and non-deployed (n = 17) participants performed a maximal effort CPET via the Bruce treadmill protocol. Indirect calorimetry and perceptual rating scales were used to measure rate of oxygen consumption ($\dot{V}O_2$), rate of carbon dioxide production ($\dot{V}CO_2$), respiratory frequency ($f_R$), tidal volume ($V_T$), minute ventilation ($\dot{V}E$), heart rate (HR), perceived exertion (RPE; 6–20 scale), and dyspnea (Borg Breathlessness Scale; 0–10 scale). A repeated measures analysis of variance (RM-ANOVA) model (2 groups: deployed vs non-deployed X 6 timepoints: 0%, 20%, 40%, 60%, 80%, and 100% $\dot{V}O_{2peak}$) was conducted for participants meeting valid effort criteria (deployed = 25; non-deployed = 11).

### Results

Significant group ($\eta^2_{partial}$ = 0.26) and interaction ($\eta^2_{partial}$ = 0.10) effects were observed such that deployed Veterans exhibited reduced $f_R$ and a greater change over time relative

authorizations did not include provisions for making individual data records publicly available, even in de-identified form. However, the authors can provide the "metadata" – i.e. the numerical (aggregated data) results used to generate the figures. Requests for access can be sent to: Towanda Smith, Privacy Officer, East Orange Campus of the VA New Jersey Healthcare System, 385 Tremont Ave East Orange, NJ 07018, PH: 973-676-1000 x201948, towanda.smith@va.gov.

**Funding:** This work was supported by Pilot Project Award # I21RX001079 from the United States (U. S.) Department of Veterans Affairs Rehabilitation Research and Development Service and supported in part by Merit Review Award # I01CX001515 and Career Development Award # IK2CX001679 from the U.S. Department of Veterans Affairs Clinical Sciences Research and Development Service. The funders had no role in study design, data collection and analysis, decision to publish, or preparation of the manuscript.

**Competing interests:** The authors have declared that no competing interests exist.

to non-deployed controls. There was also a significant group effect for dyspnea ratings ($\eta^2_{partial}$ = 0.18) showing higher values in deployed participants. Exploratory correlational analyses revealed significant associations between dyspnea ratings and $f_R$ at 80% ($R^2$ = 0.34) and 100% ($R^2$ = 0.17) of $\dot{V}O_{2peak}$, but only in deployed Veterans.

## Conclusion

Relative to non-deployed controls, Veterans deployed to SWA exhibited reduced $f_R$ and greater dyspnea during maximal exercise. Further, associations between these parameters occurred only in deployed Veterans. These findings support an association between SWA deployment and affected respiratory health, and also highlight the utility of CPET in the clinical evaluation of deployment-related dyspnea in Veterans.

## Background

Since 2001, more than three million United States military personnel have deployed to Southwest Asia and Afghanistan (SWA) in support of multiple military operations. Over this time span a considerable increase in post-deployment respiratory symptoms has been reported, prompting concerns for the respiratory health of deployed Veterans [1–5]. The National Academies of Sciences, Engineering and Medicine concluded there is sufficient evidence of an association between SWA deployment and respiratory symptoms (i.e., cough, wheeze, and dyspnea) but insufficient evidence to support an association with respiratory conditions [6]. The non-specific nature of respiratory symptoms, particularly dyspnea on exertion, is challenging to assess as it may be present in those with and without objective pulmonary function abnormalities. Although there have been reports of a spectrum of respiratory conditions following deployment [1–5], the majority of symptomatic military personnel and Veterans remain undiagnosed after evaluation and have pulmonary function within normal limits [3,5,7,8]. For these individuals whereby dyspnea seems disproportionate to function, cardiopulmonary exercise testing (CPET) may provide unique insight though has received less attention in this population.

Dyspnea is a subjective experience of breathing discomfort that is not exclusive to the respiratory system [9]. As such, it is important to thoroughly consider all potential contributing factors (e.g., deconditioning, respiratory, cardiovascular, and/or metabolic). CPET is ideally suited for this task as it provides the opportunity to both recreate symptoms on exertion as well as identify the system(s) underlying observed limitations. Given the inhalational exposures during deployment as well as the respiratory symptoms endorsed by military personnel and Veterans, greater attention to the ventilatory responses during exercise appears warranted. Determining a mechanistic role for ventilation on exertional dyspnea is best achieved by considering both lung mechanics and the ventilatory response to metabolic stimuli [10]. Ventilatory limitation to exercise has traditionally been evaluated by comparing the minute ventilation ($\dot{V}E$) at the end of exercise with the predicted or measured maximal voluntary ventilation to assess breathing reserve capacity as well as $\dot{V}E$ relative to the production of carbon dioxide ($\dot{V}E/\dot{V}CO_2$) to assess ventilatory efficiency. Analyzing CPET data with attention to respiratory mechanics and efficiency might enable a greater understanding of exertional dyspnea reported following deployment.

Despite the potential benefits of using CPET to understand dyspnea and its relevance to this population of interest, few studies have thoroughly investigated exercise ventilatory

responses and their relationship to dyspnea [1,3,7,11,12]. We have previously found that the components of exercise V̇E afforded insight into evaluating exercise intolerance among deployed veterans of earlier miliary conflicts [13]. Hence, the purpose of this study is to evaluate key CPET parameters and patterns in SWA deployed Veterans relative to non-deployed control participants. Our primary hypothesis was that symptomatic deployed Veterans without a diagnosed respiratory condition have inefficient ventilation patterns during maximal exercise that is associated with dyspnea. Testing this hypothesis would advance the understanding of CPET's ability to elucidate breathing and cardiorespiratory patterns that are reflective of deployment-related dyspnea and exercise intolerance in Veterans.

## Materials and methods

This observational, case-control pilot study (NCT01754922) involved two visits where participants completed questionnaires, provided blood samples, and completed physiological testing. On one of these visits, participants performed spirometry and underwent maximal cardiopulmonary exercise testing (CPET) which is the focus of the present analysis. All study procedures were conducted at the War Related Illness and Injury Study Center located within the East Orange, New Jersey Veterans Affairs Hospital. Participants were recruited from 09/01/13 to 10/01/15.

### Participants

Forty-eight non-treatment seeking volunteers participated in this pilot study and were assigned to groups based on their deployment history–i.e., deployed to SWA or a non-deployed control group. The deployed group consisted of Veterans deployed to SWA ($\geq$ 90 consecutive days) and the non-deployed control group consisted of similar era, but non-deployed, Veterans and civilians (i.e., non-exposure group). Participants from either group were excluded from the study if any of the following were present: 1) absolute contraindications to exercise testing [14], 2) pre-military history of asthma, 3) neurological impairment or disorder, 4) uncontrolled hypertension (SPB $>$ 160 mmHg; DBP $>$ 100 mmHg), 5) severe or moderate traumatic brain injury within the past 3 years, or 6) any contraindications for spirometry (i.e., eye, chest, or abdominal surgery in last 3 months, history of stroke, heart attack or coughing up blood in past 3 months, and history of collapsed lung or aneurysm) [9,15]. All exclusion criteria were determined via self-report and verified via electronic health record in cases where self-reported information was uncertain. Study procedures were reviewed and approved by the VA New Jersey Health Care System Institutional Review Board (IRB #01193). All participants provided written informed consent before initiating study procedures.

### Questionnaires

Participants completed multiple questionnaires, including a detailed medical screening, to assess overall health and any respiratory issues. As part of the medical screening, physical activity levels (mins/day and days/week) were calculated using the short of version of the International Physical Activity Questionnaire [16]. Functional limitation questions were used to rate participant difficulty to achieve listed activities. Ratings were based on a 0–6 scale where 0 = "I don't know", 6 = "I don't do this activity", and where 1–5 ranges from "not at all difficult" to "can't do it at all". A rating of 3 "somewhat difficult" to 5 "can't do it at all" was considered to be functionally limited. The Veterans version of the Short Form 36 health survey was used to examine physical (physical component summary; PCS) and mental (mental component summary; MCS) health. The normalized national average for the PCS and MCS scores is 50 with a standard deviation of 10 [17]. To measure dyspnea and other respiratory symptoms and

their effect on day-to-day life, the St. George's Respiratory Questionnaire (SGRQ) was used. In brief, this questionnaire contains four components: 1) frequency and severity of symptoms, 2) activities that cause or are limited by breathlessness, 3) an impact component on how respiratory symptoms affect day-to-day life, and 4) a total score considering all the other parts. Each section is graded from 0 to 100 with higher scores indicating greater limitations [18,19]. Self-reported environmental exposures during deployment were assessed using the US Army Public Health Command Deployment Respiratory Air Respiratory Exposures Questionnaire. As seen in our prior work [8], Veterans deployed to the SWA were asked to rate the frequency, duration, and intensity of potential exposures via a 0–4 point Likert-type scale. Exposures included 1) sand and dust, 2) smoke from burning trash, 3) exhaust and diesel fumes, 4) industrial air pollution.

## Spirometry

All participants performed spirometry maneuvers prior to CPET in accordance with recommended guidelines [20,21], using a flow-sensing device (Cosmed Quark PFT; Rome, Italy) which was calibrated before each use. Spirometric variables are presented in their actual units and expressed as a percent of predicted [22] and included: forced vital capacity (FVC), forced expiratory volume ($FEV_1$), and their ratio ($FEV_1/FVC$). Abnormal results were defined as observed values of $FEV_1/FVC$ below the lower limit of normal as defined by Hankinson et al. [22].

## Cardiopulmonary exercise testing

Participants performed a maximal effort CPET on a treadmill (Trackmaster) using the Bruce Protocol until volitional exhaustion [23]. Heart rate and rhythm (Cosmed T12x; Rome, Italy), as well as oxygen saturation were continuously monitored. Blood pressure was manually auscultated every 3 min during exercise and every 2 min into recovery. Perceived exertion (RPE; 6–20 scale) and dyspnea (Borg Breathlessness Scale; 0–10 scale) were measured every 2 min throughout exercise and at min 2-, 5- and 10 of recovery [24]. Pulmonary gas exchange and ventilation were measured breath-by-breath via an oronasal face mask interfaced with a metabolic cart (Cosmed Quark CPET).

A clinical exercise physiologist supervised all CPETs and ensured participant safety. Testing was terminated when end criteria were reached as judged by the test administrator or when participants were no longer able to maintain speed and grade despite verbal encouragement. Valid effort was defined as meeting two or more of the following criteria: 1) peak respiratory exchange ratio (RER) $\geq$ 1.1, 2) peak heart rate $\geq$ 85% of age-predicted maximum, 3) no change in the rate of oxygen consumption ($\dot{V}O_2$) < 2.1 ml·min·kg$^{-1}$ over last min ($\dot{V}O_2$ plateau), 4) RPE rating of $\geq$ 17, and/or 5) blood lactate level reaching sex- and age-related thresholds [25,26]. $\dot{V}O_2$ plateau was considered present if at least two of three clinical exercise physiologists who independently examined the CPET report ruled that a valid plateau was achieved.

## Exercise data processing

Raw breath-by-breath data were visually inspected and averaged (30 sec. intervals) for offline analysis in MATLAB (v20.0, Mathworks; Natick, MA). Respiratory compensation point and ventilatory anaerobic threshold (VAT) were determined by lab personnel using the $\dot{V}E/\dot{V}CO_2$ plot and the modified V-slope approach, respectively [27]. Most prioritized variables were directly measured during CPET which included rate of oxygen consumption ($\dot{V}O_2$), rate of

carbon dioxide production ($\dot{V}CO_2$), respiratory frequency ($f_R$), tidal volume ($V_T$), minute ventilation ($\dot{V}E$), respiratory exchange ratio (RER) and heart rate (HR). Since dyspnea and RPE were measured every 2 min during testing, these variables were exported individually by lab personnel based on the closest rating to target time. For example, if a subject reached 60% of their $\dot{V}O_{2peak}$ during minute 4:30 of exercise, the dyspnea and RPE rating given at min 4 was used. Dyspnea and RPE ratings are only presented as % $\dot{V}O_{2peak}$.

Breathing reserve (BR) and ventilatory efficiency ($\dot{V}E/\dot{V}CO_2$) were calculated to assess ventilatory limitations. BR was calculated using predicted maximal voluntary ventilation (MVV = $FEV_1$ x 40; L/min) where BR = $(1 - \dot{V}E_{peak}/MVV)$X 100) with values $\leq$ 15% considered abnormal [28]. Ventilatory efficiency was defined as the regression slope relating $\dot{V}E$ to $\dot{V}CO_2$ slope from the start of exercise to respiratory compensation point if applicable [29]. $\dot{V}E/\dot{V}CO_2$ slope values $\geq$ 35 were considered abnormal.

### Statistical analysis

Statistical analyses were conducted in SPSS 26 (IBM Corp., Armonk, NY). All analyses were limited only to those participants who met criteria for valid effort. Group differences in participant characteristics, pulmonary function, and ventilatory limitations (BR, $\dot{V}E/\dot{V}CO_2$, $\dot{V}E/MVV$) were analyzed using independent *t*-tests and Hedges' *d* (*d*) effect sizes. Effect sizes of 0.25, 0.50, and 0.80 were interpreted as small, medium, and large, respectively. Normality was checked using the Kolmogorov-Smirnov test and non-normal data were analyzed using Mann-Whitney U. Chi-square or Fisher's exact test was used to determine differences in groups when comparing categorical data.

For our primary analysis, separate repeated measures analysis of variance models (2 groups: deployed vs non-deployed at 6 relative intensities: 0%, 20%, 40%, 60%, 80%, and 100% $\dot{V}CO_{2peak}$) were used to analyze select CPET variables ($\dot{V}O_2$, $\dot{V}CO_2$, $\dot{V}E$, $V_T$, $f_R$, dyspnea, RPE, RER, HR). Degrees of freedom were adjusted (Greenhouse-Geisser) when the sphericity assumption was violated (Mauchly's test of Sphericity). The magnitude of main and interaction effects was assessed with *F*-statistics and partial eta squared effect size ($\eta^2_{partial}$). Values of 0.01, 0.06, and 0.14 for $\eta^2_{partial}$ were interpreted as small, medium, and large effects respectively.

## Results

### Participant characteristics

All 48 participants completed the CPET protocol. Of the 48 participants, 36 participants met criteria for valid effort (25 deployed, 11 non-deployed). The remaining 12 participants were excluded from the primary analytic sample (Fig 1). Demographics, self-reported health, and physical activity are reported in Table 1 (Fisher's exact for sex, P = 1.00). The only significant between-group difference was the VR-36 PCS score (P = 0.048). There was moderate between-group difference characterized by worse physical health in the deployed group relative to the non-deployed group (*d* = -0.75; 95% CI: -1.48, -0.01). Self-reported respiratory symptoms are presented in Table 2. Deployed Veterans reported significantly worse respiratory health across all four categories examined (P<0.05) with effect size differences ranging from *d* = 0.98–2.02. Self-reported exposures are reported in Table 3. Regarding functional limitations, Fisher's exact test indicated that the proportion of participants reporting limited function (rating of 3 or higher) did not significantly differ between groups—climbing upstairs: 16% vs 0%,

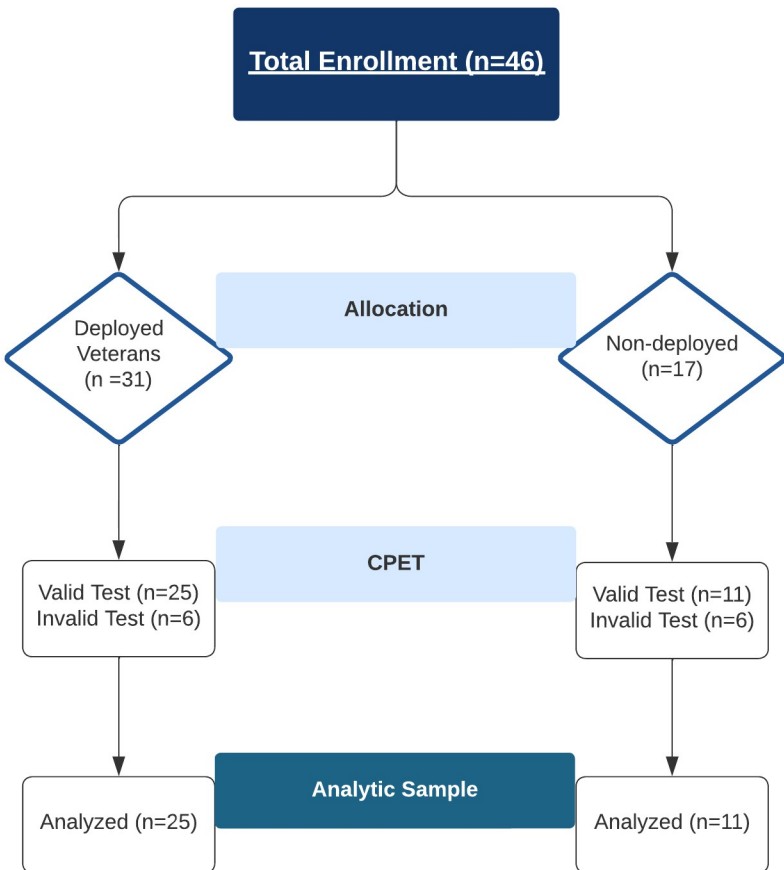

**Fig 1. Flow chart of study enrollment, testing, and data analysis for deployed (n = 25) and non-deployed (n = 11) groups.**

P = 0.290; walk up a hill: 16% vs 0%, P = 0.290; Running 1 mile: 16% vs 9.1%, P = 1.00; walk 1 mile: 8% vs 0%, P = 1.00; walk ¼ mile: 0% vs 0%, P = 1.00).

## Spirometry

Results from spirometry testing are presented in Table 4. Significant differences (P<0.05) were observed for the percent predicted for FVC ($d$ = -1.02; 95% CI: -1.79, -0.26) and FEV$_1$ ($d$ = -1.00; 95% CI: -1.76, -0.23). Among deployed Veterans 23% had FEV$_1$/FVC ratios below the LLN while only 9% for the non-deployed group (Fisher's Exact Test P = 0.223).

## Cardiopulmonary exercise testing

From the repeated measure tests, $f_R$ showed both a significant group (P = 0.001; $\eta^2_{partial}$ = 0.26) and interaction (P = 0.022; $\eta^2_{partial}$ = 0.10) effect. There was a significant group effect for dyspnea ratings (P = 0.011; $\eta^2_{partial}$ = 0.18). Group and interaction effects were not observed for any other CPET parameter, but all variables had a significant time-effect (all P<0.001). Fig 2 shows mean values for $\dot{V}E$, $V_T$, and $f_R$ expressed as a percentage of $\dot{V}O_{2peak}$. Results from the repeated measures models are presented in Table 5. Mean (SD) for prioritized CPET parameters at each stage (i.e., rest, VAT, peak exercise) and expressed as a percentage of $\dot{V}O_{2peak}$ are reported in Tables 6 & 7.

**Table 1. Comparison of demographics and self-reported physical activity and health between deployed (n = 25) and non-deployed (n = 11) participants.**

| | Deployed | Non-deployed | Effect size (95% CI) | P-value |
|---|---|---|---|---|
| **Age (years)** | 36.40 (7.89) | 31.09 (8.20) | 0.65 (-0.06, 1.36) | 0.075 |
| **Sex (male/female)** | 20/5 | 9/2 | N/A | 1.0 |
| **Body mass index (kg/m$^2$)** | 28.63 (5.46) | 26.78 (3.42) | 0.36 (-0.34, 1.06) | 0.310 |
| **Smoking status** | | | | |
| Current/Former/Never | 4/6/15 | 0/4/7 | N/A | |
| Pack-Years | 1.54 (3.80) | .47 (0.95) | 0.32 (-0.38, 1.02) | 0.367 |
| **Deployment length (months)** | 16.36 (7.25) | N/A | N/A | |
| **Race/Ethnicity** | | | | |
| White/Non-Hispanic | 10 | 2 | N/A | |
| White/Hispanic | 7 | 1 | N/A | |
| Black/Non-Hispanic | 6 | 8 | N/A | |
| Asian | 2 | 0 | N/A | |
| **Physical activity (Total min/week)** | 180.40 (205.46) | 227.73 (198.30) | -0.23 (-0.92, 0.47) | 0.212 |
| **Physical health (VR-36 PCS)$^\pm$** | 48.79 (8.95) | 55.37 (7.48)* | -0.75 (-1.48, -0.01) | 0.048 |
| **Mental health (VR-36 MCS)** | 41.49 (13.35) | 48.32 (17.61)* | -0.46 (-1.18, 0.27) | 0.288 |

**Note.** Continuous data are presented as mean (SD) and categorical data are presented as frequency counts; VR-36 MCS, Veterans RAND 36-Item Health Survey Mental Component Score; VR-36 PCS, Veterans RAND 36-Item Health Survey Physical Component Score.

*Missing data (n = 1).

## Ventilatory limitations

BR and $\dot{V}E/\dot{V}CO_2$ slope results are presented in Table 8. Neither variable was significantly different between groups. Fisher's exact test revealed that the proportion of participants exhibiting abnormal BR did not differ between deployed Veterans and non-deployed controls (4%/18%; P = 0.223). Similarly, no group differences were observed for abnormal $\dot{V}E/\dot{V}CO_2$ (8%/0%; P = 1.00).

## Discussion

A growing recognition of the austere environmental conditions of SWA and deployment-related exposures (e.g., vapors, gases, dusts and fumes) has led to several studies exploring respiratory complaints of deployed Veterans [2–5,30]. Many of these studies are unable to identify a clear etiology for these complaints which we hypothesized may be due, in part, to the nature of resting pulmonary function testing and that assessments on exertion (i.e., CPET)

**Table 2. Mean (SD) comparison of self-reported respiratory symptoms between deployed (n = 25) and non-deployed (n = 11) participants.**

| | Deployed | Non-deployed | Effect size (95% CI) | P-value |
|---|---|---|---|---|
| **SGRQ** | | | | |
| Symptom$^\pm$ | 28.53 (12.23)* | 6.53 (5.54) | 2.02 (1.16, 2.88) | <0.001 |
| Activity$^\pm$ | 21.65 (16.91) | 2.74 (5.66) | 1.27 (0.51, 2.04) | <0.001 |
| Impact$^\pm$ | 9.35 (8.85) | 1.67 (3.75) | 0.98 (0.23, 1.72) | 0.012 |
| Total$^\pm$ | 16.30 (10.04)* | 2.79 (3.46) | 1.54 (0.84, 2.34) | <0.001 |

**Note.** SGRQ, St. George's Respiratory Questionnaire.

$^\pm$ Significant between-group difference (α = 0.05).

*Missing data (n = 1).

**Table 3. Frequency of self-reported deployment exposures among deployed Veterans (n = 25).**

| Exposure characteristic | Sand & dust | Smoke from burn pits | Vehicle exhaust | Regional air pollution |
|---|---|---|---|---|
| **Frequency** | | | | |
| Not exposed | 0 (0%) | 0 (0%) | 1 (4%) | 9 (36%) |
| Seldom to few days | 0 (0%) | 5 (20%) | 1 (4%) | 5 (20%) |
| Occasionally to half-time | 3 (12%) | 3 (12%) | 2 (8%) | 3 (12%) |
| Majority of days | 6 (24%) | 8 (32%) | 6 (24%) | 3 (12%) |
| Daily | 16 (64%) | 9 (36%) | 15 (60%) | 5 (20%) |
| **Duration (hours)** | | | | |
| Not exposed | 0 (0%) | 0 (0%) | 1 (4%) | 9 (36%) |
| 0–3 | 2 (8%) | 7 (28%) | 3 (12%) | 5 (20%) |
| 4–12 | 8 (32%) | 6 (24%) | 6 (24%) | 7 (28%) |
| 13–20 | 6 (24%) | 7 (28%) | 7 (28%) | 1 (4%) |
| >20 | 9 (36%) | 5 (20% | 8 (32%) | 3 (12%) |
| **Average Intensity** | | | | |
| Not exposed | 0 (0%) | 0 (0%) | 1 (4%) | 9 (36%) |
| No Noticeable effects | 3 (12%) | 3 (12%) | 3 (12%) | 6 (24%) |
| Mild health effects | 8 (32%) | 11 (44%) | 14 (56%) | 7 (28%) |
| Moderate Health Effects | 13 (52%) | 10 (40%) | 6 (24%) | 3 (12%) |
| Severe Health Effects | 1 (4%) | 1 (4%) | 1 (4%) | 0 (0%) |

may yield greater insight. In the present study, our deployed Veterans endorsed considerable exposures (Table 3) and substantially greater respiratory limitations (Table 2) than non-deployed controls despite not seeking treatment for their symptoms. Spirometric evaluation did demonstrate greater obstruction and CPET revealed alterations in breathing patterns among deployed Veterans. These findings are discussed in greater detail below.

## Larger between-group differences were observed for $f_R$ and dyspnea relative to other CPET indices

The primary aim of this study was to evaluate the potential utility of CPET to provide physiological insight into respiratory symptoms reported by Veterans deployed to SWA (i.e.,

**Table 4. Mean (SD) comparison of spirometry indices between deployed (n = 25) and non-deployed (n = 11) participants.**

| | Deployed | Non-deployed | Effect size (95% CI) | P-value |
|---|---|---|---|---|
| **Spirometry** | | | | |
| FVC | 4.66 (1.07)* | 4.92 (0.94) | -0.03 (-0.75, 0.70) | 0.488 |
| FVC Percent Predicted ± | 99.90 (14.41)* | 114.18 (14.84) | -1.02 (-1.79, -0.26) | 0.012 |
| FEV$_1$ | 3.61 (0.74)* | 4.01 (0.81) | -0.51 (-1.25, 0.22) | 0.157 |
| FEV$_1$ Percent Predicted ± | 95.20 (16.29)* | 111.64 (17.61) | -1.00 (-1.76, -0.23) | 0.012 |
| FEV$_1$/FVC | 77.54 (7.50)* | 82.29 (5.47) | -0.67 (-1.41, 0.07) | 0.072 |
| FEV$_1$/FVC Percent Predicted | 95.10 (9.53)* | 98.18 (6.46) | -0.33 (-1.06, 0.40) | 0.342 |
| PEF | 8.16 (1.83)* | 9.63 (2.25) | -0.73 (-1.47, 0.02) | 0.053 |
| FEV$_1$/FVC < LLN | 5 (22.7%)* | 1 (9.1%) | N/A | 0.223 |

**Note.** FVC, forced vital capacity; FEV$_1$, forced expiratory volume in one second; LLN, lower limit of normal.

± Significant between-group difference (α = 0.05).

*Missing data (n = 3).

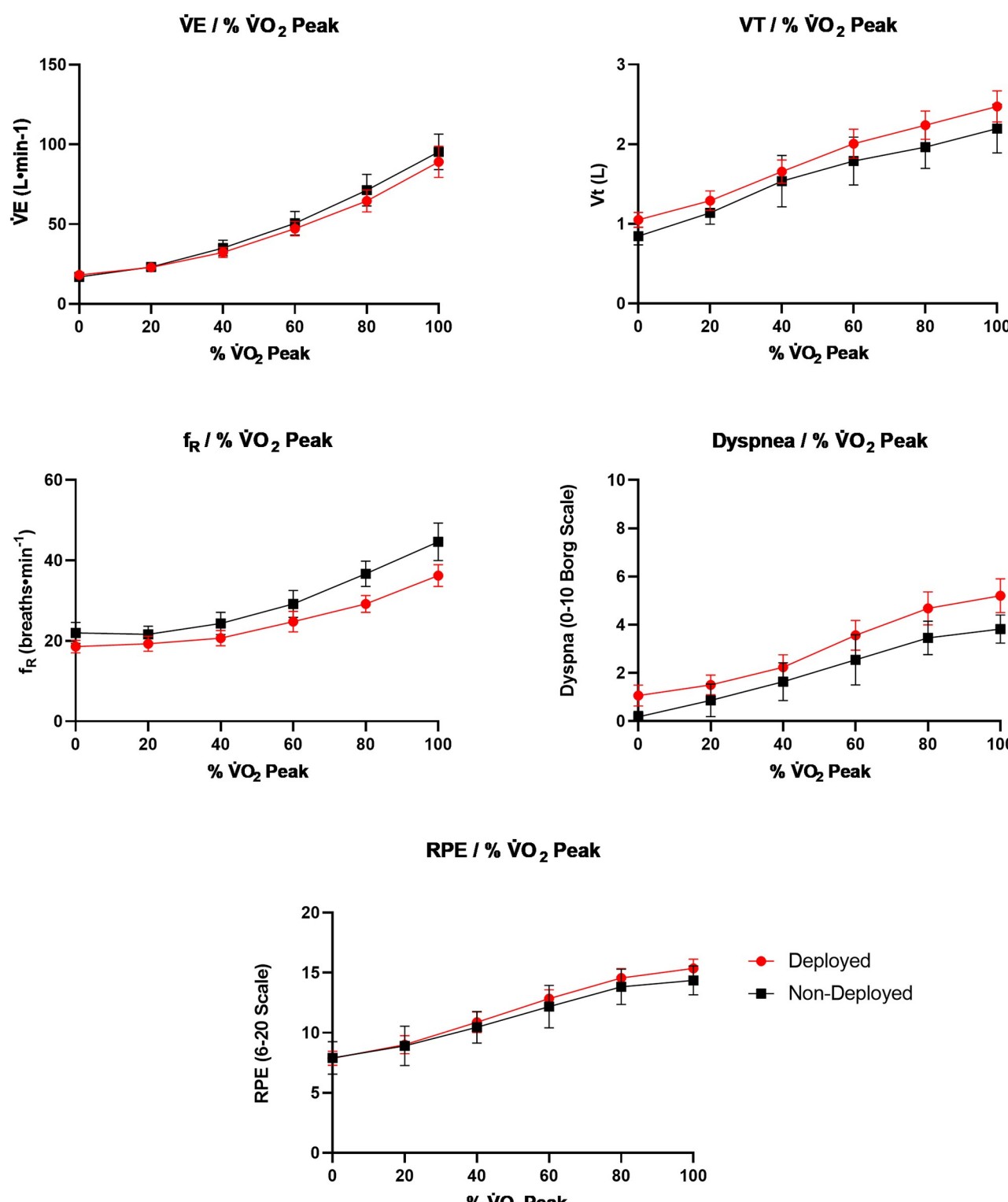

**Fig 2. Mean values for ventilatory variables, dyspnea ratings, and ratings of perceived exertion at 0, 20, 40, 60, 80, and 100% of peak oxygen consumption for deployed (n = 25) and non-deployed (n = 11) groups.**

**Table 5. Results of separate repeated measures analysis of variance models comparing select cardiopulmonary exercise testing parameters between deployed (n = 25) and non-deployed (n = 11) participants.**

| Variables | Interaction effect | Group effect | Time effect |
|---|---|---|---|
| | F-Score \| Partial Eta$^2$ \| P-Value | F-Score \| Partial Eta$^2$\| P-Value | F-Score \| Partial Eta$^2$\| P-Value |
| $\dot{V}O_2$ (mL) | 1.760 \| 0.049 \|0.194 | 0.365 \| 0.011 \|0.550 | 374.117 \| 0.917$^{\pm}$ \|<0.001 |
| $\dot{V}CO_2$ (mL) | 0.917 \| 0.026 \|0.363 | 0.467 \| 0.140 \|0.499 | 394.652 \| 0.921$^{\pm}$ \|<0.001 |
| $\dot{V}E$ (L·min$^{-1}$) | 1.108 \| 0.032 \|0.317 | 0.700 \| 0.020 \|0.409 | 363.468 \| 0.914$^{\pm}$ \|<0.001 |
| $V_T$ (L) | 0.714 \| 0.021 \|0.503 | 2.968 \| 0.080 \|0.094 | 202.003 \| 0.856$^{\pm}$ \|<0.001 |
| $f_R$ (breaths·min$^{-1}$) | 3.623 \| 0.096$^{\pm}$ \|0.022 | 12.030 \| 0.261$^{\pm}$ \|0.001 | 159.157 \| 0.824$^{\pm}$ \|<0.001 |
| RER | 0.324 \| 0.009 \|0.830 | 0.202 \| 0.006 \|0.656 | 319.767 \| 0.904$^{\pm}$ \|<0.001 |
| Dyspnea | 0.842 \| 0.024 \|0.485 | 7.329 \| 0.180$^{\pm}$ \|0.011 | 85.010 \| 0.714$^{\pm}$ \|<0.001 |
| Rating of perceived exertion | 0.718 \| 0.021 \|0.584 | 0.740 \| 0.021 \|0.396 | 146.113 \| 0.811$^{\pm}$ \|<0.001 |

**Note.** $\dot{V}O_2$, Oxygen consumption; $\dot{V}CO_2$, carbon dioxide production; $f_R$, respiratory frequency; $V_T$, tidal volume; $\dot{V}E$, minute ventilation; RER, respiratory exchange ratio.
$^{\pm}$ Significant effect (α = 0.05).

dyspnea). Specifically, we tested whether SWA deployment is associated with deficiencies in cardiorespiratory health by comparing physiological and perceptual responses to maximal exercise between deployed Veterans and non-deployed controls. After restricting our analytic dataset to 36 participants (25 deployed, 11 non deployed) meeting criteria for a valid peak effort (Fig 1), we observed minimal between-group differences for a majority of CPET variables ($\dot{V}E$, $\dot{V}CO_2$, $\dot{V}O_2$, $V_T$, RER, and HR). However, we observed moderate-large group ($\eta^2_{partial}$ = 0.26) and group-by-time interaction ($\eta^2_{partial}$ = 0.096) effects for $f_R$. As shown in Fig 2, these findings are characterized by slower $f_R$ and a greater change over time in deployed Veterans relative to non-deployed controls. Additionally, a large group effect was observed for exertional dyspnea such that deployed Veterans exhibited higher ratings than non-deployed

**Table 6. Mean (SD) values for ventilatory variables and dyspnea ratings for between deployed (n = 25) and non-deployed participants (n = 11) at 0, 20, 40, 60, 80, and 100% of peak oxygen consumption.**

| | $\dot{V}E$ (L·min$^{-1}$) | | $V_T$ (L) | | $f_R$ (breaths·min$^{-1}$) | | Dyspnea (0–10) | |
|---|---|---|---|---|---|---|---|---|
| | Deployed | ES (95% CI) | Deployed | ES (95% CI) | Deployed | ES (95% CI) | Deployed | ES (95% CI) |
| | Non-deployed | | Non-deployed | | Non-deployed | | Non-deployed | |
| **Rest** | 18.13 (3.74) | 0.36 (-0.34, 1.06) | 1.05 (0.23)$^{\pm}$ | 0.96 (0.22, 1.68) | 18.59 (3.79)$^{\pm}$ | -0.86 (-1.58, -0.13) | 1.06 (1.05)$^{\pm}$ | 0.95 (0.21, 1.67) |
| | 16.90 (2.14) | | 0.85 (0.17)$^{\pm}$ | | 21.95 (3.93)$^{\pm}$ | | 0.18 (0.34)$^{\pm}$ | |
| **20%** | 22.91 (4.41) | -0.06 (-0.75, 0.63) | 1.29 (0.30) | 0.54 (-0.17, 1.24) | 19.27 (4.46) | -0.57 (-1.27, 0.14) | 1.50 (1.00) | 0.62 (-0.09, 1.33) |
| | 23.16 (3.25) | | 1.14 (0.21) | | 21.64 (2.98) | | 0.86 (1.00) | |
| **40%** | 32.32 (7.55) | -0.36 (-1.05, 0.34) | 1.66 (0.35) | 0.30 (-0.40, 0.99) | 20.65 (4.67)$^{\pm}$ | -0.80 (-1.51,-0.07) | 2.24 (1.26) | 0.48 (-0.23, 1.18) |
| | 35.06 (7.32) | | 1.54 (0.48) | | 24.32 (4.08)$^{\pm}$ | | 1.64 (1.16) | |
| **60%** | 47.12 (10.94) | -0.30 (-0.99, 0.40) | 2.01 (0.44) | 0.47 (-0.23, 1.17) | 24.80 (6.18) | -0.73 (-1.44, -0.01) | 3.56 (1.50) | 0.65 (-0.06, 1.36) |
| | 50.52 (11.07) | | 1.79 (0.45) | | 29.15 (5.00) | | 2.55 (1.56) | |
| **80%** | 64.55 (16.82) | -0.40 (-1.10, 0.30) | 2.24 (0.43) | 0.63 (-0.08, 1.33) | 29.19 (5.10) | -1.47 (-2.24, -0.68) | 4.68 (1.65)$^{\pm}$ | 0.80 (0.08, 1.51) |
| | 71.27 (14.77) | | 1.96 (0.40) | | 36.67 (4.67) | | 3.45 (1.04)$^{\pm}$ | |
| **100%** | 89.09 (23.57) | -0.28 (-0.98, 0.42) | 2.47 (0.47) | 0.58 (-0.14, 1.28) | 36.21 (6.53) | -1.23 (-1.98, -0.47) | 5.20 (1.71)$^{\pm}$ | 0.89 (0.16, 1.61) |
| | 95.33 (16.51) | | 2.20 (0.46) | | 44.61 (6.95) | | 3.82 (0.87)$^{\pm}$ | |

**Note.** ES, effect size; $f_R$, respiratory frequency; $V_T$, tidal volume; $\dot{V}E$, minute ventilation.
$^{\pm}$ Significant between-group difference (α = 0.05).

**Table 7. Mean (SD) comparison of values for select cardiopulmonary exercise testing parameters at rest, ventilatory anaerobic threshold, and peak exercise between deployed (n = 25) and non-deployed (n = 11) participants.**

| | Rest | | | Ventilatory anaerobic threshold | | | Peak | | |
|---|---|---|---|---|---|---|---|---|---|
| Variables | Deployed | Non-deployed | P-Value | Deployed | Non-deployed | P-Value | Deployed | Non-deployed | P-Value |
| $\dot{V}O_2$ (ml·min·kg) | 4.82 (1.39) | 5.50 (1.61) | 0.226 | 21.49 (4.15) | 26.82 (9.48) | 0.098 | 34.68 (8.94) | 34.60 (7.60) | 0.114 |
| $\dot{V}CO_2$ (mL) | 310.70 (68.18) | 323.46 (76.22) | 0.310 | 1116.68 (404.45) | 1205.33 (964.68) | 0.697 | 3288.15 (830.50) | 3553.64 (771.73) | 0.374 |
| $\dot{V}E$ (L·min$^{-1}$) | 11.50 (2.55) | 11.77 (2.25) | 0.381 | 30.49 (10.21) | 33.48 (24.50) | 0.606 | 93.88 (24.55) | 100.27 (17.83) | 0.443 |
| $V_T$ (L) | 0.81 (0.25) | 0.71 (0.19) | 0.246 | 1.63 (0.46) | 1.34 (0.43) | 0.090 | 2.47 (0.48) | 2.24 (0.51) | 0.206 |
| $f_R$ (breaths·min$^{-1}$) | 15.52 (2.74)$^{\pm}$ | 18.54 (3.94)$^{\pm}$ | 0.012 | 19.71 (6.4) | 24.68 (0.15) | 0.092 | 38.30 (7.16)$^{\pm}$ | 46.18 (9.37)$^{\pm}$ | 0.009 |
| Heart rate (beats·min$^{-1}$) | 73.35 (11.62) | 67.01 (13.66) | 0.162 | 126.52 (16.71) | 133.55 (16.11) | 0.249 | 172.33 (13.87) | 170.62 (11.89) | 0.725 |
| RER | 0.80 (0.08) | 0.77 (0.06) | 0.258 | 0.74 (0.08) | 0.74 (0.15) | 0.882 | 1.14 (0.50) | 1.14 (0.07) | 0.953 |
| Dyspnea (0–10) | 1.06 (1.05)$^{\pm}$ | 0.18 (0.34)$^{\pm}$ | 0.011 | N/A | N/A | N/A | 5.20 (1.70)$^{\pm}$ | 3.82 (0.87)$^{\pm}$ | 0.016 |
| Perceived exertion (6–20) | 7.89 (0.08) | 7.90 (2.02) | 0.961 | N/A | N/A | N/A | 15.36 (1.87) | 14.36 (1.80) | 0.146 |

**Note.** $\dot{V}O_2$, Oxygen consumption; $\dot{V}CO_2$, carbon dioxide production; $f_R$, respiratory frequency; $V_T$, tidal volume; $\dot{V}E$, minute ventilation; RER, respiratory exchange ratio.

$^{\pm}$ Significant between-group difference ($\alpha$ = 0.05).

controls ($\eta^2_{partial}$ = 0.18). Furthermore, the findings do not appear to be confounded by ventilatory limitations, as indicated by non-significant group differences in BR (Table 8).

It is noteworthy that spirometry results showed limitations (FEV$_1$/FVC < LLN) for 5 deployed Veterans (~23%) and 1 non-deployed control (~9%). To ensure that any observed effects from our study were not related to potential restrictive or obstructive lung patterns, all analyses were repeated excluding participants who had FEV$_1$/FVC < LLN. Overall, similar findings were observed in the restricted dataset (Tables 9–15). Despite limiting the dataset, deployed Veterans showed distinct breathing patterns where they employed deeper but slower breaths during exercise. These differences were more visible at greater exercise intensities. In our estimation, these collective symptom and pulmonary function results indicate that ventilatory mechanical limitations do not explain the respiratory limitations experienced by our sample of deployed Veterans.

Our findings can be interpreted in the context of prior research involving both healthy adults and deployment-related illnesses such as Gulf War Illness (GWI). Consistent with earlier work which found that $V_T$ is determined more by metabolic factors than $f_R$ [31–34], a study of healthy adults found that $f_R$, but not $V_T$, responds rapidly to changes in workload during high-intensity cycling and recovery, is independent from metabolic factors ($\dot{V}CO_2$, $\dot{V}O_2$),

**Table 8. Mean (SD) comparison of ventilatory limitations between deployed (n = 25) and non-deployed (n = 11) participants.**

| | Deployed | Non-deployed | Effect size | P-Value |
|---|---|---|---|---|
| **BR** | 34.03 (11.50)* | 36.24 (12.68) | -0.18 (-0.89, 0.53) | 0.617 |
| **BR < 15%** | 1 (4%)* | 2 (18.18%) | N/A | 0.223 |
| **VE/VCO$_2$ slope** | 28.49 (3.47) | 28.35 (2.37) | 0.04 (-0.65, 0.74) | 0.906 |
| **VE/VCO$_2$ > 35%** | 2 (8%) | 0 (0%) | N/A | 1.0 |

**Note.** BR, Breathing reserve; $\dot{V}E/\dot{V}CO_2$, ventilatory efficiency.

*Missing data (n = 3).

**Table 9. Comparison of demographics and self-reported physical activity and health between deployed (n = 17) and non-deployed (n = 10) participants from the limited dataset.**

| | Deployed | Non-deployed | Effect size (95% CI) | P-Value |
|---|---|---|---|---|
| **Age (years)** ± | 38.94 (7.89) | 30.70 (8.54) | -0.98 (-1.78, -0.17) | 0.18 |
| **Sex (male/female)** | 13/4 | 8/2 | N/A | N/A |
| **Body mass index (kg/m²)** | 28.00 (5.44) | 27.25 (3.22) | -0.15 (-0.91, 0.61) | .700 |
| **Smoking status** | | | | |
| Current/Former/Never | 4/2/11 | 0/4/6 | N/A | N/A |
| Pack-Years | 1.54 (3.80) | 0.47 (0.95) | -0.41 (-1.17, 0.36) | 0.304 |
| **Deployment length (months)** | 17.56 (8.46) | N/A | N/A | N/A |
| **Race/Ethnicity** | | | | |
| White/Non-Hispanic | 6 | 1 | N/A | N/A |
| White/Hispanic | 4 | 1 | N/A | N/A |
| Black/Non-Hispanic | 5 | 8 | N/A | N/A |
| Asian | 2 | 0 | N/A | N/A |
| **Physical activity (Total min/week)** | 152.35 (185.44) | 250.50 (193.27) | 0.51 (-0.27, 1.27) | 0.203 |
| **Physical health (VR-36 PCS)** ± | 47.64 (8.39) | 55.14 (7.9)* | 0.88 (0.05, 1.70) | 0.037 |
| **Mental health (VR-36 MCS)** | 40.06 (15.36) | 47.86 (18.62)* | 0.46 (-.34, 1.25) | 0.263 |

**Note.** Continuous data are presented as mean (SD) and categorical data are presented as frequency counts; VR-36 MCS, Veterans RAND 36-Item Health Survey Mental Component Score; VR-36 PCS, Veterans RAND 36-Item Health Survey Physical Component Score.

± Significant between-group difference (α = 0.05).

*Missing data (n = 1).

and is strongly associated with RPE [35]. Further, our prior study of Veterans with GWI found altered breathing patterns in Veterans with GWI, as indicated by higher $V_T$ and lower $f_R$ compared to non-symptomatic controls [13]. Additionally, group differences in $f_R$ were larger than $V_T$, suggesting that $\dot{V}E$ was primarily driven by $f_R$ in symptomatic Veterans. Integrating these prior studies with the present study, our observation of slower $f_R$ in deployed Veterans may represent a centrally-mediated, learned strategy to mitigate dyspnea symptoms during exercise (i.e., breathing slower may help decrease feelings of breathlessness). However, it is important to note that we did not collect data on breathing patterns prior to development of dyspnea symptoms, so longitudinal CPET studies assessing pre- and post-deployment health are clearly needed to establish temporal sequence.

**Table 10. Mean (SD) comparison of self-reported respiratory symptoms between deployed (n = 17) and non-deployed (n = 11) participants from the limited dataset.**

| | Deployed | Non-deployed | Effect size (95% CI) | P-Value |
|---|---|---|---|---|
| **SGRQ** | | | | |
| Symptom± | 28.67 (11.96)* | 6.55 (5.84) | 2.11 (1.14, 3.05) | 0.001 |
| Activity± | 22.11 (17.37) | 2.42 (5.85) | 1.33 (0.48, 2.16) | <0.001 |
| Impact± | 9.38 (9.81) | 1.84 (3.90) | 0.89 (0.09, 1.68) | 0.002 |
| Total± | 16.65 (10.74)* | 2.76 (3.64) | 1.57 (0.66, 2.45) | 0.029 |

**Note.** SGRQ, St. George's Respiratory Questionnaire.

± Significant between-group difference (α = 0.05).

*Missing data (n = 1).

**Table 11. Frequency of self-reported deployment exposures among deployed Veterans (n = 17).**

| Exposure characteristic | Sand & dust | Smoke from burn pits | Vehicle exhaust | Regional air pollution |
|---|---|---|---|---|
| **Frequency** | | | | |
| Not exposed | 0 (0%) | 0 (0%) | 0 (0%) | 7 (41.2%) |
| Seldom to few days | 0 (0%) | 3 (17.6%) | 1 (5.9%) | 2 (11.8%) |
| Occasionally to half-time | 2 (11.8%) | 3 (17.6%) | 1 (5.9%) | 2 (11.8%) |
| Majority of days | 5 (29.4%) | 5 (29.4%) | 5 (29.4%) | 2 (11.8%) |
| Daily | 10 (58.8%) | 6 (35.3%) | 10 (58.8%) | 4 (23.5%) |
| **Duration (hours)** | | | | |
| Not exposed | 0 (0%) | 0 (0%) | 0 (0%) | 7 (41.2%) |
| 0–3 | 2 (11.8%) | 4 (23.5%) | 2 (11.8%) | 2 (11.8%) |
| 4–12 | 5 (29.4%) | 4 (23.5%) | 5 (29.4%) | 5 (29.4%) |
| 13–20 | 4 (23.5%) | 4 (23.5%) | 3 (17.6%) | 0 (0%) |
| >20 | 6 (35.3%) | 5 (29.4%) | 7 (41.2%) | 3 (17.6%) |
| **Average Intensity** | | | | |
| Not exposed | 0 (0%) | 0 (0%) | 0 (0%) | 7 (41.2%) |
| No noticeable effects | 2 (11.8%) | 2 (11.8%) | 3 (17.6%) | 1 (5.9%) |
| Mild health effects | 5 (29.4%) | 7 (41.2%) | 9 (52.9%) | 6 (35.3%) |
| Moderate health effects | 9 (52.9%) | 7 (41.2%) | 4 (23.5%) | 3 (17.6%) |
| Severe health effects | 1 (5.9%) | 1 (5.9%) | 1 (5.9%) | 0 (0%) |

## Respiratory frequency was positively associated with dyspnea ratings in deployed Veterans

Following our observation of differential $f_R$ patterns between deployed and non-deployed participants, we conducted exploratory correlational analyses which revealed that dyspnea ratings were significantly associated with $f_R$ at 80% (r = 0.58) and 100% (r = 0.41) of $\dot{V}O_{2peak}$, but only in deployed Veterans (Figs 3 and 4). The design of the present study does not lend itself to determining whether there is a mechanistic link between $f_R$ and dyspnea, but we are not the only group to observe an association between these two variables. For instance, in healthy young adults who completed a staged cycle ergometry protocol (50W/4min), Tsukada

**Table 12. Results of separate repeated measures analysis of variance models comparing select cardiopulmonary exercise testing parameters between deployed (n = 17) and non-deployed (n = 10) participants from the limited dataset.**

| Variables | Interaction effect | Group effect | Time effect |
|---|---|---|---|
| | F-Score \| Partial Eta$^2$\| P-Value | F-Score \| Partial Eta$^2$\| P-Value | F-Score \| Partial Eta$^2$\| P-Value |
| $\dot{V}O_2$ (mL) | 4.325 \| 0.147$^\pm$\| 0.097 | 2.403 \| 0.088\|0.798 | 313.80 \| 0.926$^\pm$\|<0.001 |
| $\dot{V}CO_2$ (mL) | 4.161 \| 0.143$^\pm$\| 0.041 | 2.466 \| 0.090\|0.129 | 368.703 \| 0.937$^\pm$\|<0.001 |
| $\dot{V}E$ (L·min$^{-1}$) | 3.260 \| 0.064\|0.064 | 2.267 \| 0.083\| 0.145 | 311.053 \| 0.926$^\pm$\|<0.001 |
| $V_T$ (L) | 0.523 \| 0.020\| 0.636 | 0.455 \| 0.018\| 0.506 | 150.973 \| 0.858$^\pm$\|<0.001 |
| $f_R$ (breaths·min$^{-1}$) | 4.735 \| 0.159$^\pm$\| 0.007 | 8.618 \| 0.256$^\pm$\|0.007 | 119.275 \| 0.827$^\pm$\|<0.001 |
| RER | 0.356 \| 0.014\|0.825 | 0.138 \| 0.005\|0.714 | 239.865 \| 0.906$^\pm$\|<0.001 |
| Dyspnea | 0.323 \|0.013\| 0.013 | 5.215 \| 0.173$^\pm$\|0.031 | 76.348 \| 0.753$^\pm$\|<0.001 |
| Rating of perceived exertion | 0.252 \| 0.010\|0.909 | 1.090 \| 0.042\|0.306 | 126.596 \| 0.835$^\pm$\|<0.001 |

**Note.** $\dot{V}O_2$, Oxygen consumption; $\dot{V}CO_2$, carbon dioxide production; $f_R$, respiratory frequency; $V_T$, tidal volume; $\dot{V}E$, minute ventilation; RER, respiratory exchange ratio.

$^\pm$ Significant effect (α = 0.05).

**Table 13. Mean (SD) values for ventilatory variables and dyspnea ratings for between deployed (n = 17) and non-deployed participants (n = 10) from the limited dataset at 0, 20, 40, 60, 80, and 100% of peak oxygen consumption.**

| | $\dot{V}E$ (L·min$^{-1}$) | | $V_T$ (L) | | $f_R$ (breaths·min$^{-1}$) | | Dyspnea (0–10) | |
|---|---|---|---|---|---|---|---|---|
| | Deployed / Non-deployed | Effect size (95% CI) | Deployed / Non-deployed | Effect size (95% CI) | Deployed / Non-deployed | Effect size (95% CI) | Deployed / Non-deployed | Effect size (95% CI) |
| **Rest** | 17.91 (3.98) | 0.29 (-0.47, 1.05) | 0.96 (0.18) | 0.74 (-0.05, 1.51) | 19.69 (3.68) | -0.69 (-1.47, 0.10) | 1.00(1.07)$^\pm$ | 1.00 (0.19, 1.80) |
| | 16.87 (2.26) | | 0.83 (0.16) | | 22.36 (3.89) | | 0.10 (0.21)$^\pm$ | |
| **20%** | 22.45 (4.47) | -0.16 (-0.92, 0.60) | 1.16 (0.23) | 0.18 (-0.58, 0.93) | 20.70 (4.60) | -0.29 (-1.05, 0.47) | 1.44 (1.06) | 0.65 (-0.13, 1.42) |
| | 23.13 (3.42) | | 1.12 (0.21) | | 21.92 (2.97) | | 0.75 (0.98) | |
| **40%** | 31.26 (6.81) | -0.50 (-1.26, 0.28) | 1.52(0.31) | -0.02 (-0.77, 0.74) | 21.73 (4.90) | -0.57 (-1.34, 0.21) | 2.29 (1.44) | 0.58 (-0.20, 1.35) |
| | 34.92 (7.70) | | 1.53 (0.51) | | 24.47 (4.27) | | 1.50 (1.13) | |
| **60%** | 44.96 (8.98) | -0.56 (-1.32, 0.22) | 1.82 (0.39) | 0.12 (-0.64, 0.87) | 26.21 (6.30) | -0.56 (-1.33, 0.22) | 3.59 (1.77) | 0.61 (-0.17, 1.38) |
| | 50.70 (11.66) | | 1.77 (0.47) | | 29.62 (5.01) | | 2.50 (1.63) | |
| **80%** | 60.69 (15.20) | -0.69 (-1.46, 0.10) | 2.07 (0.40) | 0.34 (-0.43, 1.10) | 29.68 (5.21)$^\pm$ | -1.55 (-2.36, -0.63) | 4.71(1.72) | 0.77 (-0.03, 1.55) |
| | 71.56 (15.54) | | 1.93 (0.41) | | 37.33 (4.35)$^\pm$ | | 3.50 (1.08) | |
| **100%** | 82.50 (21.42) | -0.64 (-1.41, 0.14) | 2.28 (0.39) | 0.28 (-.48, 1.04) | 36.40 (6.68)$^\pm$ | -1.35 (-2.18, -0.50) | 5.06 (1.75)$^\pm$ | 0.75 (-0.04, 1.53) |
| | 87.43 (20.72) | | 2.16 (0.46) | | 45.60 (6.46)$^\pm$ | | 3.90 (0.88)$^\pm$ | |

**Note.** ES, effect size; $f_R$, respiratory frequency; $V_T$, tidal volume; $\dot{V}E$, minute ventilation.

$^\pm$ Significant between-group difference ($\alpha = 0.05$).

**Table 14. Mean (SD) comparison of values for select cardiopulmonary exercise testing parameters at rest, ventilatory anaerobic threshold, and peak exercise between deployed (n = 17) and non-deployed (n = 10) participants from the limited dataset.**

| | Rest | | | Ventilatory anaerobic threshold | | | Peak | | |
|---|---|---|---|---|---|---|---|---|---|
| Variables | Deployed | Non-deployed | P-Value | Deployed | Non-deployed | P-Value | Deployed | Non-deployed | P-Value |
| $\dot{V}O_2$ (ml·min·kg) | 4.80 (1.55) | 5.43 (1.68) | 0.333 | 16.96 (4.23) | 19.66 (8.53) | 0.28 | 34.59 (8.68) | 39.96 (9.37) | 0.144 |
| $\dot{V}CO_2$ (mL) | 38.18 (4.31) | 37.80 (5.12) | 0.254 | 985.42 (326.68) | 1256.92 (1000.74) | 0.308 | 3029.52 (683.08) | 3607.57 (791.32) | 0.056 |
| $\dot{V}E$ (L·min$^{-1}$) | 10.68 (2.33) | 11.63 (2.33) | 0.332 | 27.71 (9.40) | 34.53 (25.56) | 0.326 | 87.27 (21.77) | 100.37 (18.79) | 0.129 |
| $V_T$ (L) | 0.71 (0.18) | 0.67 (0.17) | 0.628 | 1.40 (0.35) | 1.35 (0.45) | 0.76 | 2.28 (0.37) | 2.20 (0.51) | 0.634 |
| $f_R$ (breaths·min$^{-1}$) | 15.99 (2.76) | 18.92 (3.94) | 0.032 | 20.77 (7.25) | 25.30 (3.49) | 0.208 | 38.59 (7.51) | 47.10 (9.33) | 0.015 |
| Heart rate (beats·min$^{-1}$) | 72.76 (12.41) | 66.72 (14.37) | 0.260 | 124.82 (16.24) | 134.20 (16.83) | 0.165 | 169.96 (13.11) | 168.96 (11.11) | 0.842 |
| RER | 0.80 (0.06) | 0.76 (0.07) | 0.161 | 0.74 (0.07) | 0.74 (0.16) | 0.90 | 1.13 (0.05) | 1.14 (0.07) | 0.794 |
| Dyspnea (0–10) | 1.00 (1.07) | 0.10 (0.21) | 0.016 | N/A | N/A | N/A | 5.06 (1.75) | 3.90 (0.86) | 0.063 |
| Perceived exertion (6–20) | 7.82 (1.01) | 7.50 (1.58) | 0.521 | N/A | N/A | N/A | 15.12 (2.06) | 14.20 (1.81) | 0.254 |

**Note.** $\dot{V}O_2$, Oxygen consumption; $\dot{V}CO_2$, carbon dioxide production; $f_R$, respiratory frequency; $V_T$, tidal volume; $\dot{V}E$, minute ventilation; RER, respiratory exchange ratio.

$^\pm$ Significant between-group difference ($\alpha = 0.05$).

**Table 15. Mean (SD) comparison of ventilatory limitations between deployed (n = 17) and non-deployed (n = 11) participants from the limited dataset.**

| | Deployed | Non-deployed | Effect size | P-Value |
|---|---|---|---|---|
| **BR** | 37.62% (9.85) | 35.78% (13.26) | -0.16 (-0.92, 0.60) | 0.684 |
| **BR < 15%** | 0 (0%) | 2 (20.0%) | N/A | N/A |
| **VE/VCO$_2$ slope** | 28.78 (3.51) | 27.94 (2.03) | -0.27 (-1.03, 0.50) | 0.495 |
| **VE/VCO$_2$ > 35%** | 2 (11.3%) | 0 (0%) | N/A | N/A |

**Note.** BR, Breathing reserve; $\dot{V}E/\dot{V}CO_2$, ventilatory efficiency.

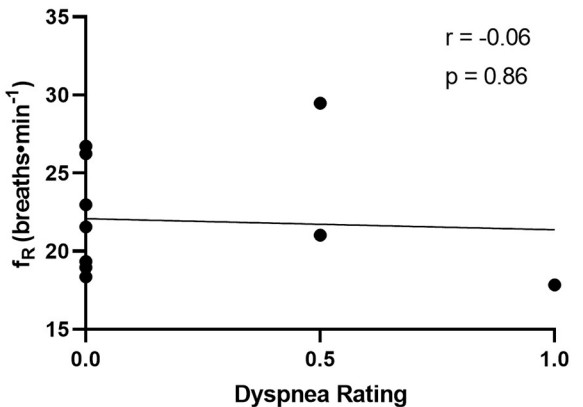

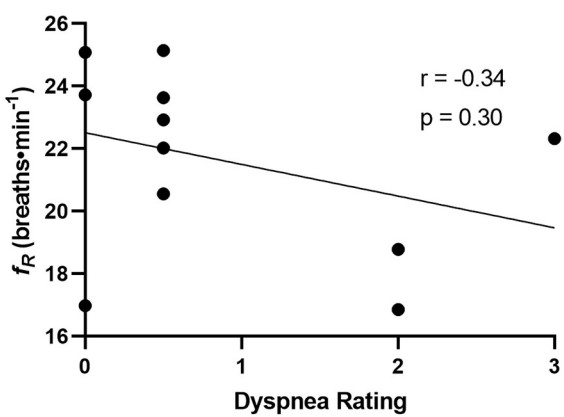

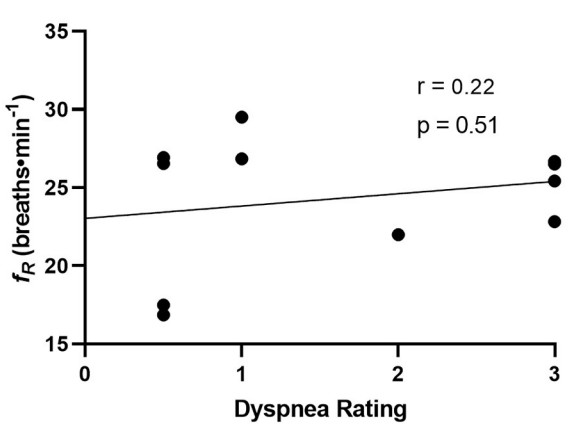

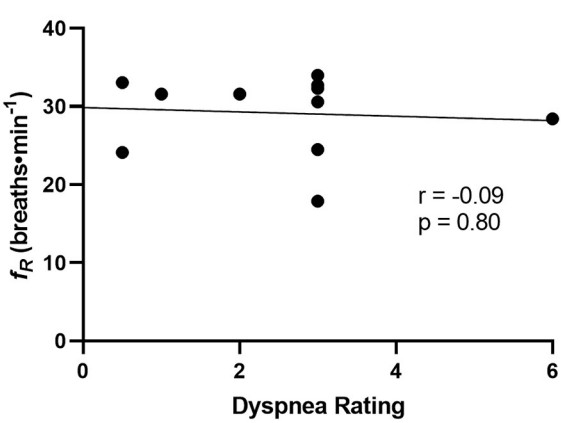

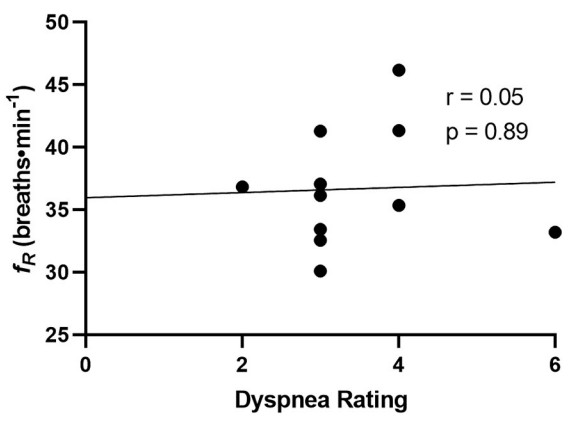

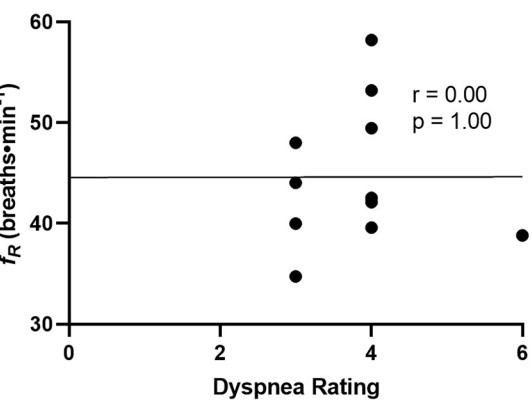

**Fig 3. Exploratory Pearson correlations (α = 0.05) between respiratory frequency and dyspnea ratings at 0, 20, 40, 60, 80, and 100% of peak oxygen consumption for the non-deployed group (n = 11).**

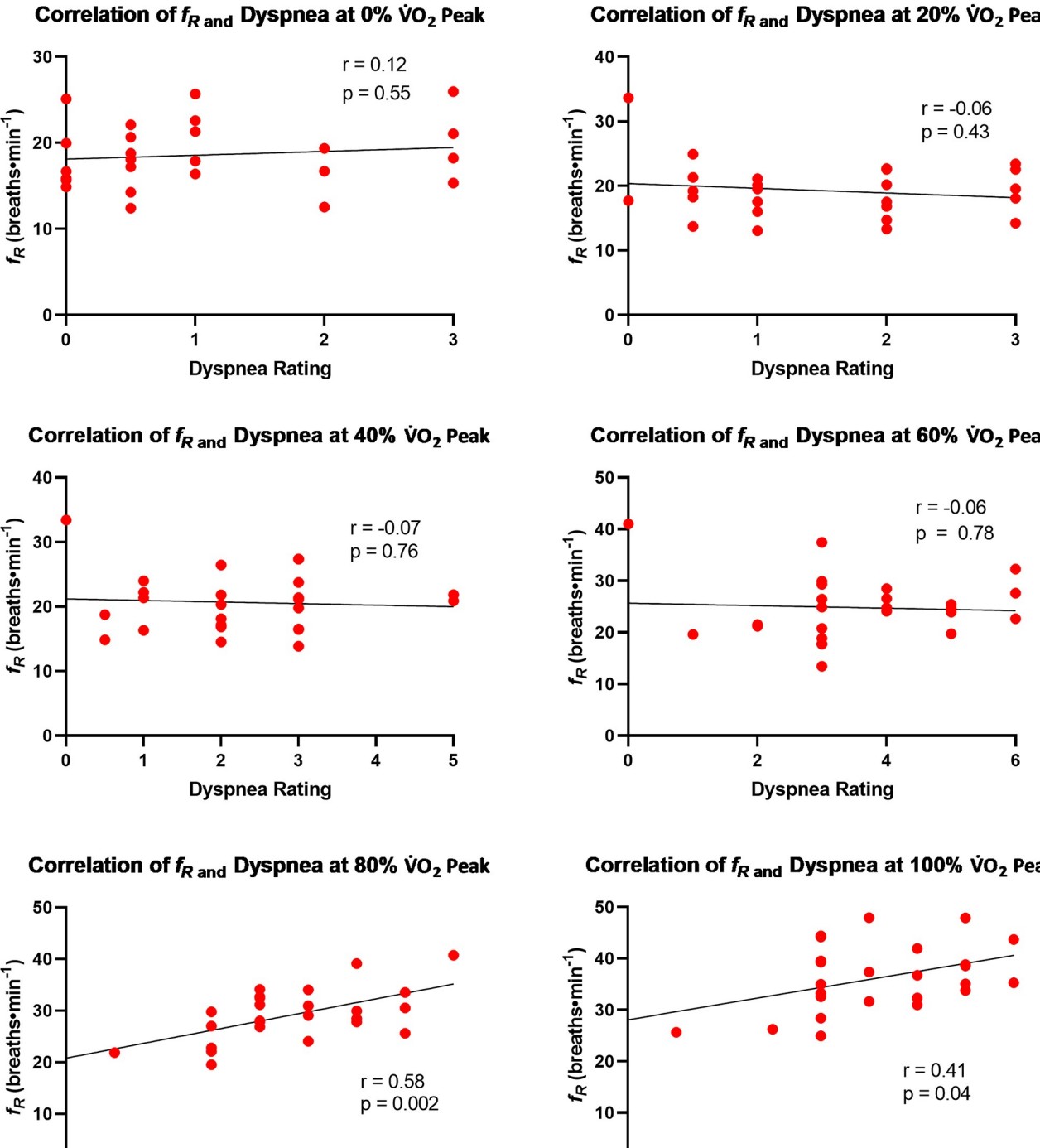

**Fig 4. Exploratory Pearson correlations ($\alpha = 0.05$) between respiratory frequency and dyspnea ratings at 0, 20, 40, 60, 80, and 100% of peak oxygen consumption for the deployed group (n = 25).**

reported that the threshold at which $f_R$ becomes tachypneic is preceded by and associated with the point at which exertional dyspnea begins to rapidly increase [36]. The investigators speculated that unpleasantness accompanied by dyspnea reaches a level which induces emotional respiratory reactions to stimulate a tachypneic breathing pattern. However, our findings are somewhat counterfactual to that interpretation. That is, despite reporting higher dyspnea ratings throughout the CPET, deployed Veterans had slower $f_R$ values than non-deployed Veterans at every timepoint (Table 6, Fig 2), as also observed in our prior work involving Veterans with Gulf War Illness [13]. Given that Tsukada focused on healthy adults whereas we focused on Veterans with deployment related exposures, perhaps people who experience significant respiratory symptoms on a day-to-day basis are less susceptible to dyspnea-driven increases in $f_R$ because they are more familiar with the experience of dyspnea than otherwise healthy people (Table 2). This hypothesis could be tested by studying the effect of experimentally manipulated dyspnea on $f_R$ in a direct comparison of healthy and symptomatic Veterans.

## Prior reporting of CPET indices in post-911 SWA Veterans is limited

Multiple other studies have conducted CPET in deployed SWA military personnel and Veterans [1,3,7,11,12]; however, analyses have focused primarily on traditional parameters (e.g., peak exercise) and do not consider dynamic exercise ventilation patterns. For instance, two separate studies by Morris and colleagues report numerous CPET values but limited to two timepoints: $\dot{V}O_{2peak}$ and VAT [3,7]. Interestingly, authors observed increased peak $f_R$ among those with dyspnea relative to controls (Mean (SD): 50.2 (12.4) vs 44.5 (6.7)) but it is unclear whether $f_R$ differences persisted at submaximal intensities. Moreover, other studies that have included CPET in their analyses [1] only presented $\dot{V}O_{2peak}$ and percent abnormal for $\dot{V}O_{2peak}$, $\dot{V}E/MVV$, $\dot{V}E/\dot{V}CO_2$, and VAT. Similar to the present study, these previously performed studies rarely identify between-group differences when restricting CPET analyses to traditional indices. It should be noted that unlike the present sample comprised of non-treatment seeking Veterans, previously published studies in military personnel and Veterans underwent CPET as part of a clinical evaluation for respiratory complaints [1,3–5,12,37]. Although the present study sample was not referred for clinical evaluation, the deployed Veteran group endorsed considerable respiratory symptom burden (Table 2).

## Limitations and future directions

This study was not without limitations. First, it is important to acknowledge the cross-sectional nature of our design. Despite excluding participants who reported having asthma prior to deployment, we did not directly measure respiratory health in this sample prior to SWA deployment. Therefore, studies that evaluate respiratory health and CPET parameters prior to and following deployment are needed to substantiate and extend our findings. Second, in light of emerging evidence arguing for a multidimensional model of dyspnea measurement [38], it is possible that there were certain aspects of dyspnea that were not captured by our 0–10 Borg Breathlessness Scale such as an affective component of dyspnea. Given the strong relationship between $f_R$ and emotion [39], and that we found greater $f_R$ in deployed participants, future attempts to explore the relationship between dyspnea and $f_R$ may also consider measuring the affective as well as sensory components of dyspnea. Third, although we did not observe signs of ventilatory limitations via examination of BR, alternative methods such as using serial inspiratory capacity maneuvers during exercise may have revealed dynamic respiratory mechanical abnormalities not observed by examining BR alone [10]. Thus, future studies should consider using serial inspiratory capacity maneuvers in addition to BR to increase confidence that

ventilatory limitations are not contributing to dyspnea ratings [10]. Fourth, it is possible that there were smaller CPET-related differences between deployed and non-deployed participants which were not observed because of statistical power. Although we started with a larger number of participants at the outset of this study, it was important to restrict our analysis only to those participants who met valid effort criteria as our prior work involving Veterans with unexplained fatigue has shown that the ability to detect smallest real differences is sometimes affected by whether or not participants met max criteria [40]. Nevertheless, our findings should be viewed as preliminary, hypothesis generating results that warrant confirmation in a larger sample.

## Conclusion

In our sample, Veterans deployed to SWA exhibit reduced $f_R$ and greater dyspnea during maximal exercise relative to non-deployed controls. Further, $f_R$ is positively associated with dyspnea ratings at 80% and 100% of $\dot{V}O_{2peak}$ in deployed Veterans but not in non-deployed controls. These findings provide support for a potential association between deployment to SWA and cardiorespiratory health. In addition, our findings highlight the utility of incorporating CPET for the evaluation of exertional dyspnea beyond that of traditional peak indices to investigate the dynamic behavior of exercise ventilation.

## Acknowledgments

The authors would like to thank the volunteers who participated in this study as well as data collection and analysis support from Bishoy Samy and Nancy Eager. This study was registered on clinicaltrials.gov (NCT01754922). The contents do not represent the views of the U.S. Department of Veterans Affairs or the United States Government.

## Author Contributions

**Conceptualization:** Jorge M. Serrador, Michael J. Falvo.

**Data curation:** Thomas Alexander, Matthew A. Watson, Jacquelyn C. Klein-Adams, Duncan S. Ndirangu, Michael J. Falvo.

**Formal analysis:** Thomas Alexander, Michael J. Falvo, Jacob B. Lindheimer.

**Funding acquisition:** Jorge M. Serrador, Michael J. Falvo.

**Investigation:** Michael J. Falvo.

**Methodology:** Thomas Alexander, Michael J. Falvo, Jacob B. Lindheimer.

**Project administration:** Jacquelyn C. Klein-Adams, Michael J. Falvo.

**Writing – original draft:** Thomas Alexander, Matthew A. Watson, Michael J. Falvo, Jacob B. Lindheimer.

**Writing – review & editing:** Thomas Alexander, Matthew A. Watson, Jacquelyn C. Klein-Adams, Duncan S. Ndirangu, Jorge M. Serrador, Michael J. Falvo, Jacob B. Lindheimer.

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
