## [Decision Letter · Decision Letter 0]

5 Feb 2023

PONE-D-22-23425Deployed Veterans exhibit distinct respiratory patterns and greater dyspnea during maximal cardiopulmonary exercisePLOS ONE

Dear Dr. Lindheimer,

Thank you for submitting your manuscript to PLOS ONE. After careful consideration, we feel that it has merit but does not fully meet PLOS ONE’s publication criteria as it currently stands. Therefore, we invite you to submit a revised version of the manuscript that addresses the points raised during the review process.

We look forward to receiving your revised manuscript.

Kind regards,

Walid Kamal Abdelbasset, Ph.D.

Academic Editor

PLOS ONE

and https://journals.plos.org/plosone/s/file?id=ba62/PLOSOne_formatting_sample_title_authors_affiliations.pdf.

“The authors would like to thank the volunteers who participated in this study as well as data collection and analysis support from Bishoy Samy and Nancy Eager. This work was supported by Pilot Project Award # I21RX001079 from the United States (U.S.) Department of Veterans Affairs Rehabilitation Research and Development Service and supported in part by Merit Review Award # I01CX001515 and Career Development Award # IK2CX001679 from the U.S. Department of Veterans Affairs Clinical Sciences Research and Development Service. This study was registered on clinicaltrials.gov (NCT01754922). The contents do not represent the views of the U.S. Department of Veterans Affairs or the United States Government.”

“Include this sentence at the end of your statement: The funders had no role in study design, data collection and analysis, decision to publish, or preparation of the manuscript.”

Reviewers' comments:

Reviewer's Responses to Questions

**Comments to the Author**

1. Is the manuscript technically sound, and do the data support the conclusions?

Reviewer #1: Yes

Reviewer #2: Partly

2. Has the statistical analysis been performed appropriately and rigorously? 

Reviewer #1: Yes

Reviewer #2: I Don't Know

3. Have the authors made all data underlying the findings in their manuscript fully available?

Reviewer #1: Yes

Reviewer #2: No

4. Is the manuscript presented in an intelligible fashion and written in standard English?

Reviewer #1: Yes

Reviewer #2: Yes

5. Review Comments to the Author

Reviewer #1: This study shoes that compared to non-deployed controls, Veterans deployed to Southwest Asia (SWA) had reduced respiratory frequency and greater dyspnea during maximal exercise. Associations of these parameters were found only in deployed Veterans. These findings, despite some limitations recognized by the authors, provide support for an association between deployment to SWA and respiratory health and highlight the utility of respiratory frequency in the clinical evaluation of deployment-related dyspnea in Veterans. The study was well conducted and is well written.

The only changes I suggest consist in the utility of providing more information for the average readers and on data from Veterans in previous conflicts in Asia, for example in Afghanistan, which involved many countries such as the USA, various European countries, Canada and Australia

Reviewer #2: Title

Title needs to be modified. “Name the trial design” should appear in the title.

All figures aren’t clear, please repeat the figures to be more clear and readable

Line 99 (Hence, the purpose of this study) is not clear –please clarify what’s the novelty of your study –why you measure CPET for SWA deployed Veterans relative to non-deployed control (what is known and what’s unknown )

As you point out in line 107 your study was an observational pilot study so please add STROBE Checklist

Line 160 (Blood pressure was manually) how it done manually during Bruce protocol (maybe have false reading) did you consider these error during the study period

In Table 1. Participant demographics, self-reported health, and physical presented

You listed the smoking status of the participant only is this the only factor that will affect the physical health of the participants

You did not put in the inclusion or exclusion criteria so we need more focus in that point

The discussion section needs to be described scientifically. Kindly frame it along the following lines:

1-Main findings of the present study

2-Comparison with other studies

3- Implication and explanation of findings

6. PLOS authors have the option to publish the peer review history of their article (what does this mean?). If published, this will include your full peer review and any attached files.

Reviewer #1: No

Reviewer #2: No

---

## [Author Response · Author response to Decision Letter 0]

6 Mar 2023

Response to Reviewers

Thank you to the editor and reviewers for taking the time to review and provide feedback on our work. We believe that the revised manuscript has been strengthened through this process and hope that our responses to your comments are to your satisfaction.

Editor: 

and https://journals.plos.org/plosone/s/file?id=ba62/PLOSOne_formatting_sample_title_authors_affiliations.pdf.

Thank you. We have closely reviewed these guidelines and formatted our documents accordingly.

Done. Thank you.

“The authors would like to thank the volunteers who participated in this study as well as data collection and analysis support from Bishoy Samy and Nancy Eager. This work was supported by Pilot Project Award # I21RX001079 from the United States (U.S.) Department of Veterans Affairs Rehabilitation Research and Development Service and supported in part by Merit Review Award # I01CX001515 and Career Development Award # IK2CX001679 from the U.S. Department of Veterans Affairs Clinical Sciences Research and Development Service. This study was registered on clinicaltrials.gov (NCT01754922). The contents do not represent the views of the U.S. Department of Veterans Affairs or the United States Government.”

“Include this sentence at the end of your statement: The funders had no role in study design, data collection and analysis, decision to publish, or preparation of the manuscript.”

We apologize for this oversight. Funding information is now removed from the Acknowledgements section of the manuscript. Thank you for changing the online submission form on our behalf. The cover letter contains the requested language to be used in the Funding Statement.

Thank you for raising this concern. We apologize for this omission. The data availability statement is now included in the cover letter and explains the ethical/legal restrictions for why a minimal underlying dataset was not uploaded (similar to our prior publication in PLOS ONE https://doi.org/10.1371/journal.pone.0224833).

 

Reviewer #1: 

1. This study shoes that compared to non-deployed controls, Veterans deployed to Southwest Asia (SWA) had reduced respiratory frequency and greater dyspnea during maximal exercise. Associations of these parameters were found only in deployed Veterans. These findings, despite some limitations recognized by the authors, provide support for an association between deployment to SWA and respiratory health and highlight the utility of respiratory frequency in the clinical evaluation of deployment-related dyspnea in Veterans. The study was well conducted and is well written. The only changes I suggest consist in the utility of providing more information for the average readers and on data from Veterans in previous conflicts in Asia, for example in Afghanistan, which involved many countries such as the USA, various European countries, Canada and Australia

Thank you. In response to your comment about referring to earlier conflicts in this region and a related comment from reviewer 2, we have added the following to the manuscript (Lines 98-100).

“We have previously found that the components of exercise V̇E afforded insight into evaluating exercise intolerance among deployed veterans of earlier miliary conflicts (13).” 

Reviewer #2: 

1. Title needs to be modified. “Name the trial design” should appear in the title.

Thank you. The title now reads, “Deployed Veterans exhibit distinct respiratory patterns and greater dyspnea during maximal cardiopulmonary exercise: A case-control study”

The study design has also been added to the Methods section.

2. All figures aren’t clear, please repeat the figures to be more clear and readable

Thank you for this comment. We agree that the figures are blurry in the PDF that is generated in the PLOS ONE manuscript portal. However, we have ensured that they meet the journal’s requirements for resolution. Please note that clicking on the hyperlink in the top right corner of each figure page will allow you to download a higher quality image than shown in the PDF (e.g. https://www.editorialmanager.com/pone/download.aspx?id=31706934&guid=53852b6a-7b61-42ee-aad9-c4c4799c0cd4&scheme=1). We will defer to the editor for further guidance on this matter if needed. 

3. Line 99 (Hence, the purpose of this study) is not clear –please clarify what’s the novelty of your study –why you measure CPET for SWA deployed Veterans relative to non-deployed control (what is known and what’s unknown )

Thank you. We have now revised this last paragraph to succinctly clarify the empirical rationale for using CPET: 

“We have previously found that the components of exercise V̇E afforded insight into evaluating exercise intolerance among deployed veterans of earlier miliary conflicts [13].” 

4. As you point out in line 107 your study was an observational pilot study so please add STROBE Checklist

Thank you for this important point. While a majority of the applicable STROBE checklist items were already incorporated in the original submission, the revised version contains additional items that were not included (e.g., study design in the title; describing the setting, locations, and relevant dates).

5. Line 160 (Blood pressure was manually) how it done manually during Bruce protocol (maybe have false reading) did you consider these error during the study period

Our laboratory – trained clinical exercise physiologists - routinely perform manual auscultation of blood pressure throughout the exercise protocol per our safety protocols. In our experience, manual auscultation of blood pressure is less prone to error than automated machines during exercise. Blood pressure was not an outcome of interest for this study or included in analyses. Therefore, we are not concerned that error associated with blood pressure measurement will impact our results.

6. In Table 1. Participant demographics, self-reported health, and physical presented, you listed the smoking status of the participant only. Is this the only factor that will affect the physical health of the participants? You did not put in the inclusion or exclusion criteria, so we need more focus in that point.

We did not exclude for smoking but recognize this may be a factor that influences cardiorespiratory performance in this population. There are likely many factors that influence physical health and considering all potential influences on health was beyond the scope of this study. Because this study is focused on lung function, we felt that smoking status was important to consider.

7. The discussion section needs to be described scientifically. Kindly frame it along the following lines:

a. 1-Main findings of the present study

b. 2-Comparison with other studies

c. 3- Implication and explanation of findings

We agree that the above elements are important to include in a discussion section. At the same time, we feel that each element is already provided and would like to bring your attention to the specific sections in which they are located. 

Main findings: 

“The primary aim of this study was to evaluate the potential utility of CPET to provide physiological insight into respiratory symptoms reported by Veterans deployed to SWA (i.e., dyspnea). Specifically, we tested whether SWA deployment is associated with deficiencies in cardiorespiratory health by comparing physiological and perceptual responses to maximal exercise between deployed Veterans and non-deployed controls. After restricting our analytic dataset to 36 participants (25 deployed, 11 non deployed) meeting criteria for a valid peak effort (Fig 1), we observed minimal between-group differences for a majority of CPET variables (V̇E, V̇CO2, V̇O2, VT, RER, and HR). However, we observed moderate-large group (η2partial =0.26) and group-by-time interaction (η2partial =0.096) effects for fR. As shown in Fig 2, these findings are characterized by slower fR and a greater change over time in deployed Veterans relative to non-deployed controls.”

Comparison with other studies: 

“Our findings can be interpreted in the context of prior research involving both healthy adults and deployment-related illnesses such as Gulf War Illness (GWI). Consistent with earlier work which found that VT is determined more by metabolic factors than fR [31–34], a study of healthy adults found that fR, but not VT, responds rapidly to changes in workload during high-intensity cycling and recovery, is independent from metabolic factors (V̇CO2, V̇O2), and is strongly associated with RPE [35]. Further, our prior study of Veterans with GWI found altered breathing patterns in Veterans with GWI, as indicated by higher VT and lower fR compared to non-symptomatic controls [13]. Additionally, group differences in fR were larger than VT, suggesting that V̇E was primarily driven by fR in symptomatic Veterans.” 

“Following our observation of differential fR patterns between deployed and non-deployed participants, we conducted exploratory correlational analyses which revealed that dyspnea ratings were significantly associated with fR at 80% (r=0.58) and 100% (r= 0.41) of V̇O2peak, but only in deployed Veterans (Figs 3-4). The design of the present study does not lend itself to determining whether there is a mechanistic link between fR and dyspnea, but we are not the only group to observe an association between these two variables. For instance, in healthy young adults who completed a staged cycle ergometry protocol (50W/4min), Tsukada reported that the threshold at which fR becomes tachypneic is preceded by and associated with the point at which exertional dyspnea begins to rapidly increase [36].” 

“Multiple other studies have conducted CPET in deployed SWA military personnel and Veterans [1,3,7,11,12]; however, analyses have focused primarily on traditional parameters (e.g., peak exercise) and do not consider dynamic exercise ventilation patterns. For instance, two separate studies by Morris and colleagues report numerous CPET values but limited to two timepoints: V̇O2peak and VAT [3,7]. Interestingly, authors observed increased peak fR among those with dyspnea relative to controls (Mean (SD): 50.2 (12.4) vs 44.5 (6.7)) but it is unclear whether fR differences persisted at submaximal intensities. Moreover, other studies that have included CPET in their analyses [1] only presented V̇O2peak and percent abnormal for V̇O2peak, V̇E/MVV, V̇E/V̇CO2, and VAT. Similar to the present study, these previously performed studies rarely identify between-group differences when restricting CPET analyses to traditional indices. It should be noted that unlike the present sample comprised of non-treatment seeking Veterans, previously published studies in military personnel and Veterans underwent CPET as part of a clinical evaluation for respiratory complaints [1,3–5,12,37]. Although the present study sample was not referred for clinical evaluation, the deployed Veteran group endorsed considerable respiratory symptom burden (Table 2).”

Implications and explanations of findings: 

“Integrating these prior studies with the present study, our observation of slower fR in deployed Veterans may represent a centrally-mediated, learned strategy to mitigate dyspnea symptoms during exercise (i.e., breathing slower may help decrease feelings of breathlessness). However, it is important to note that we did not collect data on breathing patterns prior to development of dyspnea symptoms, so longitudinal CPET studies assessing pre- and post-deployment health are clearly needed to establish temporal sequence.”

“The investigators speculated that unpleasantness accompanied by dyspnea reaches a level which induces emotional respiratory reactions to stimulate a tachypneic breathing pattern. However, our findings are somewhat counterfactual to that interpretation. That is, despite reporting higher dyspnea ratings throughout the CPET, deployed Veterans had slower fR values than non-deployed Veterans at every timepoint (Table 6, Fig 2), as also observed in our prior work involving Veterans with Gulf War Illness [13]. Given that Tsukada focused on healthy adults whereas we focused on Veterans with deployment related exposures, perhaps people who experience significant respiratory symptoms on a day-to-day basis are less susceptible to dyspnea-driven increases in fR because they are more familiar with the experience of dyspnea than otherwise healthy people (Table 2). This hypothesis could be tested by studying the effect of experimentally manipulated dyspnea on fR in a direct comparison of healthy and symptomatic Veterans.”

“In our sample, Veterans deployed to SWA exhibit reduced fR and greater dyspnea during maximal exercise relative to non-deployed controls. Further, fR is positively associated with dyspnea ratings at 80% and 100% of V̇O2peak in deployed Veterans but not in non-deployed controls. These findings provide support for a potential association between deployment to SWA and cardiorespiratory health. In addition, our findings highlight the utility of incorporating CPET for the evaluation of exertional dyspnea beyond that of traditional peak indices to investigate the dynamic behavior of exercise ventilation.”

---

## [Decision Letter · Decision Letter 1]

2 May 2023

PONE-D-22-23425R1Deployed Veterans exhibit distinct respiratory patterns and greater dyspnea during maximal cardiopulmonary exercise: A case-control studyPLOS ONE

Dear Dr. Lindheimer,

Thank you for submitting your manuscript to PLOS ONE. After careful consideration, we feel that it has merit but does not fully meet PLOS ONE’s publication criteria as it currently stands. Therefore, we invite you to submit a revised version of the manuscript that addresses the points raised during the review process.

We look forward to receiving your revised manuscript.

Kind regards,

Kalyana Chakravarthy Bairapareddy, PhD

Academic Editor

PLOS ONE

Journal Requirements:

Additional Editor Comments:

The reviewers have recommended minor revision. It is recommended to do the following corrections and resubmit the manuscript.

1. Some references are very old (22, 23,24, 29). Please update the references.

2. Change the subheading "introduction" to "Background" in accordance with the guidelines. In the abstract, remove the hypothesis.

3. Mention the level of significance in the results.

Reviewers' comments:

Reviewer's Responses to Questions

**Comments to the Author**

1. If the authors have adequately addressed your comments raised in a previous round of review and you feel that this manuscript is now acceptable for publication, you may indicate that here to bypass the “Comments to the Author” section, enter your conflict of interest statement in the “Confidential to Editor” section, and submit your "Accept" recommendation.

Reviewer #1: (No Response)

Reviewer #2: All comments have been addressed

2. Is the manuscript technically sound, and do the data support the conclusions?

Reviewer #1: (No Response)

Reviewer #2: Yes

3. Has the statistical analysis been performed appropriately and rigorously? 

Reviewer #1: Yes

Reviewer #2: I Don't Know

4. Have the authors made all data underlying the findings in their manuscript fully available?

Reviewer #1: Yes

Reviewer #2: Yes

5. Is the manuscript presented in an intelligible fashion and written in standard English?

Reviewer #1: Yes

Reviewer #2: Yes

6. Review Comments to the Author

Reviewer #1: (No Response)

Reviewer #2: thanks a lot for your reply

7. PLOS authors have the option to publish the peer review history of their article (what does this mean?). If published, this will include your full peer review and any attached files.

Reviewer #1: No

Reviewer #2: No

---

## [Author Response · Author response to Decision Letter 1]

4 May 2023

Response to Reviewers

Thank you for the additional feedback. We are delighted to hear that the manuscript only required minor revisions at this stage. Please see our responses to your comments below.

1. Some references are very old (22, 23,24, 29). Please update the references.

If possible, we would prefer to keep these specific references as they are the original source for the corresponding method that we are describing in that section. 

2. Change the subheading "introduction" to "Background" in accordance with the guidelines. In the abstract, remove the hypothesis.

Done.

3. Mention the level of significance in the results.

Done. We have also added P-values to the tables where appropriate.

---

## [Editor Report · Decision Letter 2]

7 May 2023

Deployed Veterans exhibit distinct respiratory patterns and greater dyspnea during maximal cardiopulmonary exercise: A case-control study

PONE-D-22-23425R2

Dear Dr. Jacob B. Lindheimer

We’re pleased to inform you that your manuscript has been judged scientifically suitable for publication and will be formally accepted for publication once it meets all outstanding technical requirements.

Kind regards,

Kalyana Chakravarthy Bairapareddy, PhD

Academic Editor

PLOS ONE
---

## [Editor Report · Acceptance letter]

15 May 2023

PONE-D-22-23425R2 

Deployed Veterans exhibit distinct respiratory patterns and greater dyspnea during maximal cardiopulmonary exercise: A case-control study 

Dear Dr. Lindheimer:

I'm pleased to inform you that your manuscript has been deemed suitable for publication in PLOS ONE. Congratulations! Your manuscript is now with our production department. 

Kind regards, 

on behalf of

Dr. Kalyana Chakravarthy Bairapareddy 

Academic Editor

PLOS ONE